# Distinguishability of keystroke dynamic template

Napa Sae-Bae[1]*, Nasir Memon[2]

1 Computer Science Department, Faculty of Science, Srinakharinwirot University, Bangkok, Thailand,
2 Computer Science Department, Tandon School of Engineering, New York University, New York, NY, United States of America

* napasa@g.swu.ac.th

## Abstract

When keystroke dynamics are used for authentication, users tend to get different levels of security due to differences in the quality of their templates. This paper addresses this issue by proposing a metric to quantify the quality of keystroke dynamic templates. That is, in behavioral biometric verification, the user's templates are generally constructed using multiple enrolled samples to capture intra-user variation. This variation is then used to normalize the distance between a set of enrolled samples and a test sample. Then a normalized distance is compared against a predefined threshold value to derive a verification decision. As a result, the coverage area for accepted samples in the original space of vector representation is discrete. Therefore, users with the higher intra-user variation suffer higher false acceptance rates (FAR). This paper proposes a metric that can be used to reflect the verification performance of individual keystroke dynamic templates in terms of FAR. Specifically, the metric is derived from statistical information of user-specific feature variations, and it has a non-decreasing property when a new feature is added to a template. The experiments are performed based on two public keystroke dynamic datasets comprising of two main types of keystroke dynamics: constrained-text and free-text, namely the CMU keystroke dynamics dataset and the Web-Based Benchmark for keystroke dynamics dataset. Experimental results based on multiple classifiers demonstrate that the proposed metric can be a good indicator of the template's false acceptance rate. Thus, it can be used to enhance the security of the user authentication system based on keystroke dynamics.

**Data Availability Statement:** CMU Keystroke Dynamics - Benchmark Data available at https://www.cs.cmu.edu/~keystroke/), Web-Based Benchmark for Keystroke Dynamics available at https://www.researchgate.net/publication/

# 1 Introduction

Text passwords have long been used as an authentication factor in computer systems. As a password can be guessed or observed, the system cannot ensure that the one who types the correct password is indeed the authorized user or simply an attacker who might have obtained the password by means of guessing [1], shoulder surfing [2], or by the other side-channel information [3–5], etc. One solution to enhance the security of password-based authentication mechanisms is to use two-factor authentication systems (2FA). Here the first factor is the password itself which is based on knowledge (what you know) factor, the second factor can be either a

259497002_Web_based_keystroke_
158dynamics_dataset.

**Funding:** NS FDA-CO-2562-9854-TH National
Science and Technology 313 Development Agency
https://www.nstda.or.th/ The funders had no role in
study design, data collection and analysis, decision
to publish, or preparation of the manuscript.

**Competing interests:** The authors have declared
that no competing interests exist.

possession factor (what you have) or a biometric factor (what you are). It has been argued that a possession factor (or token) could cause security issues as it can be lost or stolen. In addition, its usability is limited as tokens need to be carried around. Therefore, the use of keystroke dynamic as a second authentication factor has been proposed to improve security without posting any extra burden on the users as it can be extracted while the password is being entered [6–9].

As a result, there has been a lot of work attempting to improve the verification performance of keystroke dynamic classifiers. However, keystroke dynamics recognition or verification performance is highly user-dependent since it is a behavioral biometric [10]. Therefore, another research direction to improve verification performance of a keystroke dynamic classifier is to predict the quality of the keystroke dynamic template in order to detect and to disqualify or remove those low quality templates from the system, as their existence has been reported in a number of previous behavioral biometric work [11, 12].

Regarding this, [13] have proposed a method to classify the quality of keystroke dynamic templates based on their verification performance. However, this metric is not designed to indicate the causes of verification inaccuracy. That is, verification errors can be classified into two types: false acceptance and false rejection. The first one is when genuine samples are falsely rejected by the system (the ratio between the number of falsely rejected genuine samples and the total number of genuine samples is called false rejection rate or FRR). The second one occurs when imposter samples are falsely accepted by the system (the ratio between the number of falsely accepted imposter samples and the total number of imposter samples is called false acceptance rate or FAR). These two types of errors cause different issues to the user. That is, the templates with high rates of FRR would cause a usability issue as the user would be rejected more often, while the templates with high rates of FAR would make the user particularly vulnerable to impersonation. As such, strategies to deal with templates with high FAR and FRR are different [14, 15]. Therefore, it is important to identify whether the problem of each low-quality template stems from a high FRR or a high FAR (or both).

This work specifically focuses on the quality index that could infer FAR of the template. That is, a method to assess the quality of keystroke dynamic templates in terms of FAR is proposed. Experiments are performed on two public datasets: the CMU keystroke dynamics dataset and the Web-Based Benchmark, comprising of two types of keystroke dynamic templates: free-text and constrain-text to demonstrate the effectiveness of the proposed method.

The rest of the paper is organized as follows. Section 3, provides background on keystroke dynamic verification algorithms used in this paper. Then, in Section 4, the proposed metric for assessing keystroke dynamic template distinctiveness is described. Next, details of the two datasets, experimental procedure, and the evaluation results, are presented in Section 5. Lastly, conclusions and possibilities for future work are discussed in Section 6.

## 2 Related work

As the number of approaches for user authentication has been proposed as an alternative to password, Bonneau et al. [16] has proposed a framework to compare and contrast those authentication schemes based on three main factors: usability, deployability, and security. In addition, the study revealed that, while traditional biometric-based authentication schemes (fingerprint, iris, voice) have some advantages over password based authentication in terms of usability (memorability, scalable-for-user, and physically-effortless) and security (physical observation and throttled-guessing), its deployability is the main setback for the schemes as it is not server and browser compatible. As such, the current use is limited to on-device authentication such as unlocking the phone or authorizing device access to use services. Consequently,

the password still plays an important role for general-purpose user authentication mechanisms. On the other hand, many proposals to resolve these issues of the original password authentication scheme have been proposed. For example, password managers that require the users to log in with their master password can be used to resolve usability in terms of memorability, scalable-for-user, physically-effortless, physical observation and but fail to resolve security in terms of physical observation and throttled-guessing. In this case, behavioral biometric gleaned from keystroke information can be used to provide resilience to physical observation and throttled-Guessing.

In addition, Qiu et al. [17] has proposed a secure three-factor authentication protocol where all three types of authentication factors are provided by a user. Moreover, Jiang et al. [18] has proposed another secure three-factor remote authentication protocol to preserve the privacy of biometric information. These proposals allow behavioral biometric gleaned from keystroke information to be used in remote setting thereby enhancing deployability of password-based two factor authentication (password string and typing pattern). As such, the mechanism can also be used for user-IoT devices authentication where data and services on these devices are becoming more sensitive and the need for secure authentication mechanism is increasing [19, 20].

## 3 Keystroke dynamic algorithm

Generally, a biometric verification system comprises of two stages: the enrollment stage where a set of biometric samples used to derive a template to later verify a user is collected, and the verification stage where a user claims an identity by providing an input biometric at the time of authentication. The system would then accept or reject the claim according to the classifier output. Details on the feature extraction methods and the classifiers used in the keystroke dynamic authentication system are described below.

### 3.1 Keystroke timing features

N-gram latency features derived from keystroke timing sequences are widely used as a feature set in keystroke dynamic recognition applications. This feature set can be applied to keystroke captured from both keyboards and touch displays (soft-keyboards) [21, 22].

Specifically, the set of n-gram features deployed in this paper was the one used in [8]. The set comprises of three types of features: an interval time (time between key-down), flight time (time between key-up and next key-down), and hold time (time between key-down and key-up) for all keys in the phrase. That is, let $Key = \{T_1^d, T_1^u, T_2^d, T_2^u, \ldots, T_n^d, T_n^u\}$ be a keystroke dynamic sample where $n$ is the number of typing characters, $T_i^d$ and $T_i^u$ are the timestamp of the $i^{th}$ keystrokes, respectively. The three types of features can be formulated as follows.

Interval time:

$$I(i) = T_{i+1}^d - T_i^d \tag{1}$$

Flight time:

$$F(i) = T_{i+1}^d - T_i^u \tag{2}$$

Hold time:

$$H(i) = T_i^u - T_i^d \tag{3}$$

## 3.2 Classifier

Once features are computed for each sample, the user template is derived from feature sets of all enrolled samples. The system then accepts the identity claimed by the user if a distance score between the enrolled samples and the input sample is less than a pre-defined threshold. The three distance functions employed in this paper are Manhattan, Euclidean, and Mahalanobis distance. Note that, as the experiment performed in this work was comprised of two datasets with fixed and varied keystroke lengths, all the keystroke feature vectors are normalized by their respective lengths before computing the distance. Details of each distance function and its template construction are as follows.

**3.2.1 Manhattan distance.** The template for this classifier consists of the mean and standard deviation of each feature derived from enrolled samples. That is, let $F = f_i$ be a feature set representing a keystroke dynamic sample. Then, the template could be defined as $T = (\mu_{f_i}, \sigma_{f_i})$, where $\mu_{f_i}$ is the mean of feature $f_i$ of the enrolled samples and $\sigma_{f_i}$ is the standard deviation of feature $f_i$ of the enrolled samples. Then the distance between template and a keystroke dynamic sample can be computed as:

$$D_{Manh}(T, F) = \sum \| w_i \times \frac{(x_i - \mu_{f_i})}{\sigma_{f_i}} \| \tag{4}$$

where $w_i = \frac{1}{N}$ and $N$ is the total number of features derived from a sample $F$ ($N = |F|$).

**3.2.2 Euclidean distance.** The template for this classifier is the same as the previous one $T = (\mu_{f_i}, \sigma_{f_i})$ where the distance between the template and a keystroke dynamic sample can be derived as:

$$D_{Eucl}(T, F) = \sqrt{\sum w_i \times \left( \frac{x_i - \mu_{f_i}}{\sigma_{f_i}} \right)^2} \tag{5}$$

**3.2.3 Mahalanobis distance.** The template for this classifier is constructed by simply concatenating feature vectors of all enrolled keystroke dynamic samples. Then the distance between the template and a keystroke dynamic sample can be derived as:

$$D_{Maha}(T, x) = (X - \mu_T)S^{-1}(X - \mu_T) \tag{6}$$

where $S$ is the covariance matrix of the user template (or concatenation of enrolled samples features), $X$ is the feature vector of a test sample, and $\mu_T$ is the mean feature vector derived from the template. Unlike the previous two, Mahalanobis distance takes into consideration the correlation between features which is more suitable when features are not independent.

## 3.3 Verification performance evaluation

Once the distance between the template and the input sample is computed, the system decides whether to accept or reject the sample. If the score is lower than the pre-defined threshold, the sample is accepted, and it is rejected otherwise. Generally, FAR (False Acceptance Rate), FAR (False Rejection Rate), and EER (Equal Error Rate) are three metrics used to evaluate the performance of the verification system. Specifically, given a pre-defined threshold, FAR is the rate at which the system incorrectly accepts imposter samples (samples from imposter users) and FRR is the rate at which the system incorrectly rejects genuine samples (samples from genuine and honest users). EER is the rate at which FAR and FRR are equal (at the corresponding threshold). Note that for the dataset that the number of genuine and forgery samples for each

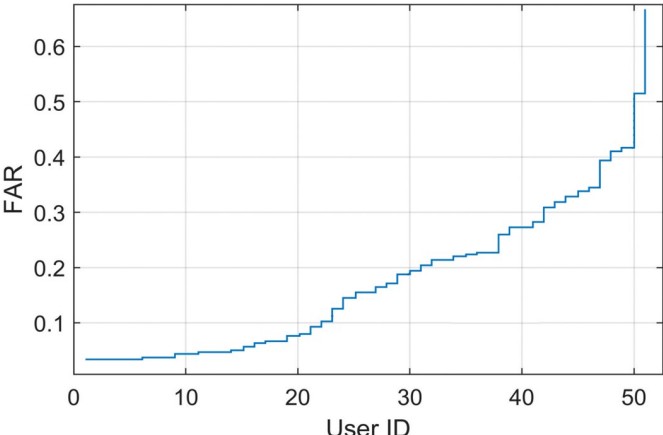

**Fig 1. A false acceptance rate of a keystroke template of each user at EER threshold according to Manhattan classifier described in [8].**

user is not equal, these rates is computed per user and the average FAR, FRR, and EER are reported.

## 4 Distinctiveness of keystroke dynamic template

As can be observed from Fig 1, the verification performance of keystroke dynamic templates in terms of the FAR show a great variation. That is, a small number of templates contribute to a large portion of the error rate in terms of FAR. In this section, a method to derive the quality of keystroke dynamic templates in terms of FAR will be described.

Typically, the distance between templates and a test sample is normalized by template feature variations [8]. As a result, feature space decreases as feature variation increases. Consequently, the template with higher feature variation would result in the higher false acceptance rate. Therefore, the inverse of feature statistical dispersion is one measure that can be used to quantify this characteristic. That is, feature distinctiveness derived $f_i$ from each feature $x_i$ is formulated as:

$$f_i = \frac{1}{\sigma_i^2} \tag{7}$$

where each feature variance $\sigma_i^2$ is defined as:

$$\sigma_i^2 = E[(x_i - \mu)^2] \tag{8}$$

In general, keystroke dynamic templates are represented by a set of features. Thus, statistical dispersion for template distinctiveness could be formulated from a multivariate distribution. Two approaches to measure statistical dispersion of multivariate distribution or total variability are total dispersion and generalized variance [23]. These two statistical dispersion measurement approaches are defined as follows.

The total dispersion or total variation is defined as the sum of all feature variances as follows:

$$\text{Total variations} = \sum_{i=1}^{n} \sigma_i^2 \tag{9}$$

The generalized variance is defined as the determinant of the covariance metric or the product of all the eigenvalues $\lambda_i$ of a feature covariance matrix as follows:

$$\text{Generalized variance} = \prod_{i=1}^{n} \lambda_i \tag{10}$$

Nevertheless, neither of the inverse forms of these two dispersion measures satisfy a monotonic property of template distinctiveness which is one important property for the proposed quality measurement to be applied on the variable-length keystroke dynamic template. That is, when another feature is added, the discrimination power of the template should not decrease. However, the inverse of total variation increases whereas, the generalized variance may decrease or increase depending on the variance of an adding feature and the correlation between an adding feature and the features currently in the set.

One way to formulate the non-decreasing function of a template distinctiveness score is to compute the inverse of feature variation individually as a feature distinctiveness. Then, the sum of these feature distinctiveness is taken as a template distinctiveness score. That is, the distinctiveness $D$ for each template can be derived from the sum of the inverse of feature variance $\sigma_i^2$ as follows:

$$D = \sum_{i=1}^{n} f_i = \sum_{i=1}^{n} \frac{1}{\sigma_i^2} \tag{11}$$

Using the proposed metric, the template with lower intra-user variation between enrolled samples (lower $\sigma$) would have higher distinctiveness as the coverage area for accepted samples in the original space of vector representation would be smaller. In addition, the keystroke template with an additional typing character would always have a higher distinctiveness. This is an important property as the distinctiveness should never decrease when additional sequences are appended to the keystroke. The effectiveness of the proposed distinctiveness score in indicating FAR will be evaluated in the following section.

## 5 Experimental result

In this section, the experimental procedures used and the results obtained are presented. Details are described below.

### 5.1 Dataset

In this paper, we used two public keystroke datasets to validate the performance of the proposed template quality score. One was the CMU Keystroke Dynamics—Benchmark Data Set (available at https://www.cs.cmu.edu/~keystroke/), and the other was the Web-Based Benchmark for Keystroke Dynamics (available at https://www.researchgate.net/publication/259497002_Web_based_keystroke_dynamics_dataset). Details of these two datasets are the following.

- The CMU keystroke dynamics dataset comprised of samples from 51 subjects. Each subject provided 400 samples from 8 sessions (50 samples per session). All samples were collected from the same typing phrase ".tie5Roanl". Note that the EER in [8] were computed per user basis and the reported performance were the average one. In addition, the imposter samples were from the first five samples of all other users instead of all 400 samples.

- The Web-Based Benchmark for keystroke dynamics dataset [24] comprised of samples from 118 subjects. Each subject provided both free-text and fixed-text keystroke samples. That is,

- free-text keystroke was where users created their own username and password phrases, and

- constraint or fixed-text keystroke was where username and password phrases were provided (the username was 'laboratoire greyc' and the password was 'S2SAME').
  All users provided samples through browsers using their own devices. In constraint or fixed-text keystroke data, the number of sessions provided by each user ranged from 1 to 47 with an average of 9.7542. In free-text keystroke data, the number of sessions provided by each user ranged from 0 to 47 with an average of 9.2034. In each session, users provided different numbers of samples. Note that not all the samples from all users were used in the experiment. First, the samples that were not typed correctly were removed. Then the samples from users that did not provide enough valid sessions were removed. Also, in order to alleviate the performance bias issue due to an imbalance number of samples from each user, the performance reported in this section was computed from the average performance of all the users (the performance numbe was computed for each user individually before computing the average).

## 5.2 Protocol

For each user, genuine samples were divided into two parts: the first $N$ samples were used as training samples to derive a user-specific template as well as the respective distinctiveness score and the rest were used as genuine test samples to evaluate the FRR of the system for that particular user. Then the FAR and FRR of the system were computed from the average FARs and FRRs from all the users. This was to prevent the user with a larger pool of (genuine and imposter) samples from having greater impact on system performance.

For the CMU dataset, training samples comprised of samples from the first four sessions (200 samples per user). Note that, in the CMU dataset, all users provided 400 genuine samples equally whereas in the Web-based Benchmark dataset the number of samples provided varies from user to user. Therefore, we limited the use of this dataset only for the user with at least four valid sessions (a valid session was a session with at least five valid samples or the samples that their keystroke were identical to the registered one) Then, all samples from that four valid sessions were used as training samples, and the rest were used as test samples. With these datasets and constraints, the performance results for three distance functions described in the previous section were at 16.00-17.16% EER for the CMU Keystroke Dynamics dataset and 13.28-25.16% average EER and for the Web-Based Benchmark Keystroke Dynamics dataset, respectively.

## 5.3 Result

In this subsection, the effectiveness of distinctiveness scores of keystroke dynamic templates are illustrated. First, the scores for all user templates were computed. Then, the templates were divided into three groups according to their distinctiveness scores. Specifically, the templates were divided into three groups according to the scores as follows,

- 'GOOD' group comprised of the templates with the highest 25th percentiles.

- 'BAD' group comprised of the templates with the lowest 25th percentiles.

- 'MID' group comprised of the rest templates.

Note that, computational overhead for deriving distinctiveness score is $O(n)$ where n is the number of features used to compute the metric (or the number of input characters). It took

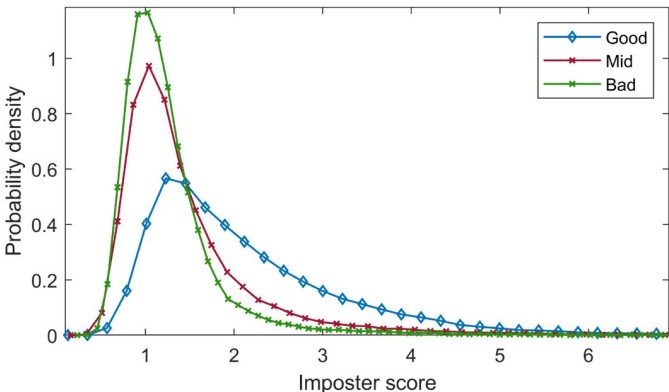

**Fig 2. Imposter score distributions of three template groups according to their distinctiveness derived from the Manhattan distance classifier.**

0.127 milliseconds on average to compute the distinctiveness from a set of keystroke samples for each template with 10 characters (or 0.0127 milliseconds per character) using Matlab R2019b running on Windows 10 with AMD Ryzen 7 3800x.

For the CMU dataset, imposter score distributions for the three groups of templates are plotted in Fig 2. A one-tailed two-sample t-test with unequal variance shows that, the mean of imposter scores of high distinctiveness templates is statistically higher than that of low distinctiveness ones, $p = (<0.001)$.

The characteristic difference in terms of FAR—decision threshold trade-off, the Spearman rank coefficient, and verification performance between each template group are demonstrated in the following.

**5.3.1 False acceptance rate.** In order to verify the effectiveness of the proposed distinctiveness scores across different types of classifier, the curves between FAR and verification threshold derived from three classifiers: Manhattan, Euclidean, and Mahalanobis distance are plotted when possible. Using these imposter scores to derive false acceptance rate for these two groups of templates across the range of verification threshold, the result in Fig 3–7 demonstrates that the FAR of high distinctiveness templates is always lower than that of low distinctiveness ones. That is, at a fixed given threshold value, the templates with low distinctiveness is more likely to accept imposter samples than the ones with high distinctiveness. The similar separation between FAR of the three groups of templates trends are observed.

Note that, for constraint keystroke in the Web-Based Benchmark for keystroke dynamics dataset, the number of training sample from the username field were not enough for constructing the Mahalanobis distance classifier. Similarly, for free-text keystroke, some users did not have enough training samples for constructing the Mahalanobis distance classifier. Therefore, only the experiments for the Manhattan and Euclidean distance were conducted in those settings.

**5.3.2 Spearman rank correlation.** The coefficients of the Spearman rank correlation between the proposed distinctiveness score and the FAR of the manhattan classifier system at the EER threshold derived from the CMU dataset is presented in Table 1 along with the performance result from classifiers employed in this study. The coefficient of -0.63 ($p < 0.001$) indicated the strong (negative) relationship between the rank of distinctiveness score and the verification performance of the templates in terms of FAR. That is, the higher the distinctiveness score implies the lower rank of FAR. Lastly, the stronger correlation found at the

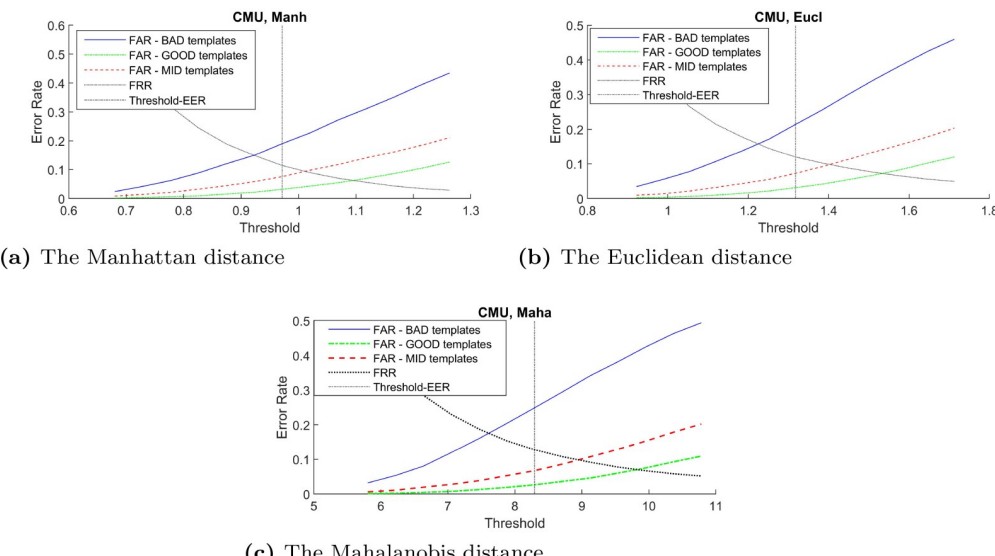

**(a)** The Manhattan distance **(b)** The Euclidean distance

**(c)** The Mahalanobis distance

**Fig 3. The CMU keystroke's performance trade-off curves between false acceptance rate and the threshold value for each template group based on three classifiers: (a) the Manhattan distance, (b) the Euclidean distance, and (c) the Mahalanobis distance.**

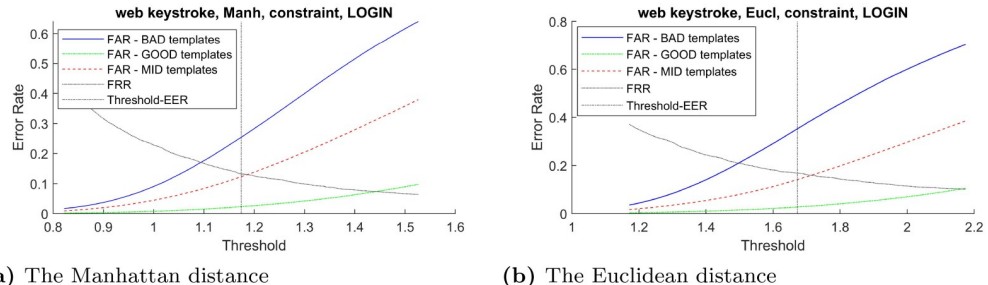

**(a)** The Manhattan distance **(b)** The Euclidean distance

**Fig 4. The Web-Based Benchmark (constraint login field -'laboratoire greyc') performance trade-off curves between false acceptance rate and the threshold value for each template group based on two classifiers: (a) the Manhattan distance and (b) the Euclidean distance.**

Euclidean and Mahalanobis distance classifiers confirmed that the proposed score can be indicative of FAR of keystroke dynamic templates on several classifiers. Similarly, the coefficient of the Spearman rank correlation between the proposed distinctiveness score and the FAR of the Manhattan classifier system at the EER threshold derived from the Web-Based Benchmark dataset is presented in Table 2.

**5.3.3 Verification performance.** As demonstrated in Figs 3–7, the templates with higher distinctiveness result in lower FAR but similar FRR when compared to the template with lower distinctiveness. This inferred that the ROC curve of templates with higher distinctiveness would be better than the one with lower distinctiveness. The FAR and FRR of these three groups of templates from the CMU dataset are presented at Table 3.

Similarly, the results from the Web-Based Benchmark dataset are presented at Table 4. Note that, the 10% FRR and 5% FRR threshold is derived at the average FRR of all users. That is the threshold is defined where the average FRR is at 10% FRR and 5%, respectively. The results clearly show that the FAR performance of the GOOD user group is much lower than

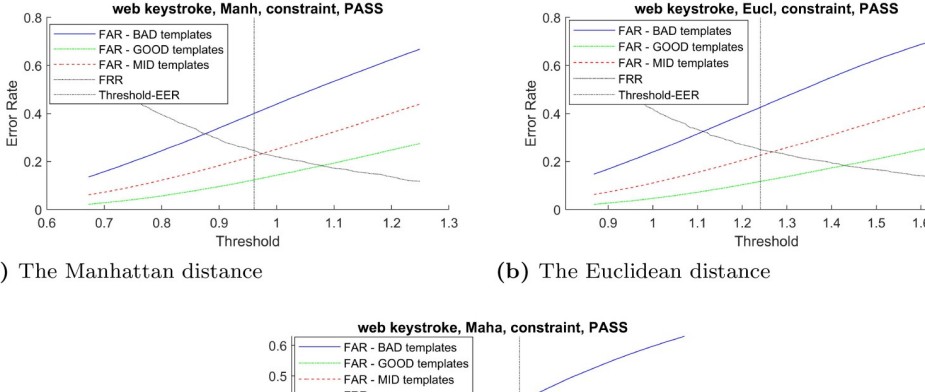

**(a)** The Manhattan distance  **(b)** The Euclidean distance

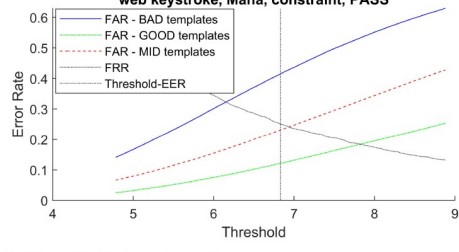

**(c)** The Mahalanobis distance

**Fig 5. The Web-Based Benchmark (constraint password -'S2SAME') performance trade-off curves between false acceptance rate and the threshold value for each template group based on three classifiers: (a) the Manhattan distance, (b) the Euclidean distance, and (c) Mahalanobis distance.**

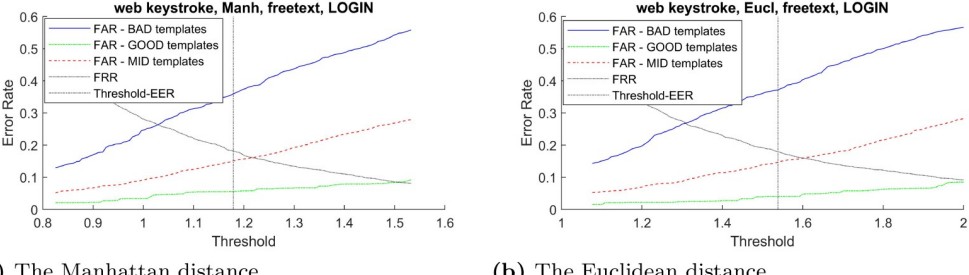

**(a)** The Manhattan distance  **(b)** The Euclidean distance

**Fig 6. The Web-Based Benchmark (free-text username) performance trade-off curves between false acceptance rate and the threshold value for each template group based on two classifiers: (a) the Manhattan distance and (b) the Euclidean distance.**

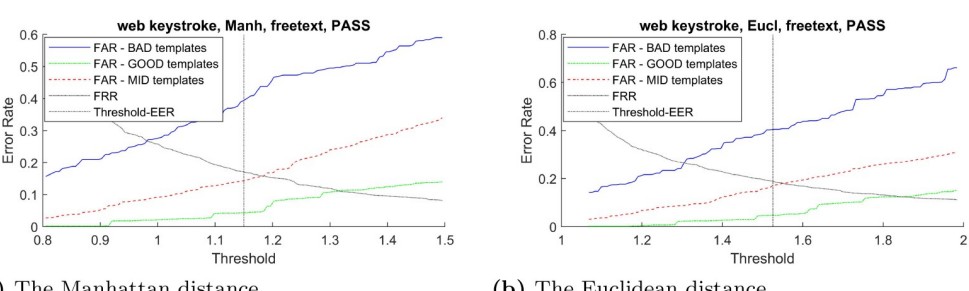

**(a)** The Manhattan distance  **(b)** The Euclidean distance

**Fig 7. The Web-Based Benchmark (free-text password field) performance trade-off curves between false acceptance rate and the threshold value for each template group based on two classifiers: (a)the Manhattan distance and (b) the Euclidean distance.**

**Table 1. The Spearman rank correlation ($\rho$) between the proposed distinctiveness score and the FAR along with the EER(%) of the system for the CMU dataset.**

| Classifier | $\rho$ | p-value | EER |
|---|---|---|---|
| Manhattan | -0.63 | 5.71e-07 | 16.00 |
| Euclidean | -0.72 | 3.22e-09 | 16.73 |
| Mahalanobis | -0.81 | 0 | 17.16 |

**Table 2. The Spearman rank correlation ($\rho$) between the proposed distinctiveness score and the FAR along with the EER(%) of the system for the Web-based keystroke dataset.**

| Field | Classifier | $\rho$ | p-value | EER |
|---|---|---|---|---|
| | | Constrain text | | |
| Username 'laboratoire greyc' | Manhattan | -0.77 | 0 | 13.28 |
| | Euclidean | -0.83 | 0 | 16.86 |
| Password 'S2SAME' | Manhattan | -0.62 | 1.65e-06 | 24.40 |
| | Euclidean | -0.70 | 2.42e-08 | 24.93 |
| | Mahalanobis | -0.66 | 2.53e-07 | 25.16 |
| | | Free-text | | |
| Username | Manhattan | -0.68 | 1.84e-08 | 18.19 |
| | Euclidean | -0.70 | 6.21e-09 | 17.94 |
| Password | Manhattan | -0.67 | 1.15e-04 | 16.99 |
| | Euclidean | -0.67 | 1.44e-04 | 18.91 |

the MID and BAD group in all three thresholds although this verification performance is still impractical for many applications.

Lastly, keystrokes of free-text username and password from the Web-Based Benchmark dataset could be combined in order to improve the verification rate. To do so, the valid pairs of correct free-text usernames and passwords must be the ones that both usernames and passwords are correctly captured (its keystroke must be identical to the registered one). Then, only

**Table 3. FAR and FRR of each template group from the CMU dataset at various threshold settings.**

| Classifier | Group | Thresholding | | | | | |
|---|---|---|---|---|---|---|---|
| | | EER | | 5% FRR | | 10% FRR | |
| | | FAR | FRR | FAR | FRR | FAR | FRR |
| Manhattan | All | 16.01% | 15.99% | 35.72% | 5.00% | 23.55% | 10.00% |
| | GOOD | 3.88% | 12.44% | 13.72% | 4.53% | 6.98% | 7.94% |
| | MID | 19.42% | 19.00% | 41.86% | 5.97% | 28.05% | 12.24% |
| | BAD | 24.74% | 16.53% | 51.57% | 4.50% | 35.61% | 9.82% |
| Euclidean | All | 16.73% | 16.74% | 46.85% | 5.00% | 28.77% | 10.00% |
| | GOOD | 3.80% | 15.12% | 20.80% | 5.21% | 9.03% | 9.53% |
| | MID | 18.52% | 19.32% | 51.23% | 5.35% | 31.64% | 11.12% |
| | BAD | 27.88% | 15.76% | 68.52% | 4.44% | 45.64% | 9.35% |
| Mahalanobis | All | 17.16% | 17.16% | 47.65% | 5.00% | 29.77% | 10.00% |
| | GOOD | 3.52% | 15.24% | 20.58% | 5.59% | 8.69% | 9.50% |
| | MID | 16.41% | 20.12% | 49.48% | 5.26% | 29.39% | 11.18% |
| | BAD | 31.56% | 16.12% | 72.90% | 4.15% | 51.21% | 9.32% |

**Table 4. FAR and FRR of each template group from the Web-Based Benchmark dataset at various threshold settings.**

| Classifier | Group | EER | | 5% FRR | | 10% FRR | |
|---|---|---|---|---|---|---|---|
| | | FAR | FRR | FAR | FRR | FAR | FRR |
| Constraint text–Username ('laboratoire greyc') | | | | | | | |
| Manhattan | All | 13.11% | 13.34% | 45.03% | 5.00% | 20.97% | 10.04% |
| | Good | 2.34% | 21.70% | 13.65% | 7.53% | 4.13% | 15.94% |
| | Mid | 12.31% | 11.31% | 46.75% | 4.34% | 20.14% | 8.70% |
| | Bad | 25.42% | 8.04% | 73.05% | 3.50% | 39.41% | 6.13% |
| Euclidean | All | 16.63% | 16.87% | 72.20% | 5.02% | 40.20% | 10.07% |
| | Good | 2.78% | 23.19% | 40.48% | 6.50% | 10.85% | 13.52% |
| | Mid | 14.17% | 15.71% | 77.94% | 4.95% | 39.34% | 9.69% |
| | Bad | 35.27% | 12.12% | 92.79% | 3.49% | 71.21% | 6.94% |
| Constraint text–Password ('S2SAME') | | | | | | | |
| Manhattan | All | 24.21% | 24.01% | 67.98% | 5.00% | 50.42% | 9.82% |
| | Good | 12.36% | 35.50% | 46.54% | 8.16% | 31.17% | 17.85% |
| | Mid | 22.28% | 21.45% | 68.53% | 4.61% | 49.15% | 7.88% |
| | Bad | 40.08% | 15.21% | 88.28% | 1.80% | 72.30% | 4.03% |
| Euclidean | All | 24.80% | 24.49% | 72.57% | 4.97% | 55.03% | 9.82% |
| | Good | 11.69% | 36.47% | 49.96% | 7.31% | 32.87% | 16.73% |
| | Mid | 22.57% | 21.85% | 74.60% | 4.50% | 53.91% | 8.04% |
| | Bad | 42.56% | 15.24% | 90.94% | 3.04% | 79.51% | 5.09% |
| Mahalanobis | All | 24.93% | 24.99% | 68.37% | 4.97% | 52.20% | 9.95% |
| | Good | 12.11% | 35.20% | 49.99% | 6.04% | 32.81% | 12.92% |
| | Mid | 23.11% | 22.41% | 70.01% | 4.88% | 52.48% | 9.67% |
| | Bad | 41.53% | 17.92% | 83.35% | 3.79% | 71.00% | 6.73% |
| Free-text –Username | | | | | | | |
| Manhattan | All | 17.86% | 18.04% | 43.27% | 4.98% | 26.88% | 9.89% |
| | Good | 5.61% | 23.92% | 16.13% | 5.16% | 7.91% | 11.92% |
| | Mid | 15.14% | 16.77% | 42.39% | 4.53% | 24.68% | 8.86% |
| | Bad | 35.96% | 14.21% | 72.31% | 5.89% | 50.57% | 10.00% |
| Euclidean | All | 17.58% | 17.80% | 49.34% | 4.96% | 28.67% | 9.90% |
| | Good | 4.07% | 23.77% | 19.15% | 4.84% | 7.31% | 12.10% |
| | Mid | 14.77% | 16.46% | 49.68% | 4.77% | 26.19% | 9.19% |
| | Bad | 37.16% | 14.03% | 78.81% | 5.56% | 55.38% | 9.02% |
| Free-text –Password) | | | | | | | |
| Manhattan | All | 16.32% | 17.87% | 51.32% | 5.45% | 27.66% | 10.24% |
| | Good | 4.32% | 18.51% | 27.15% | 4.13% | 11.91% | 9.12% |
| | Mid | 14.34% | 18.02% | 51.81% | 6.20% | 27.26% | 11.43% |
| | Bad | 39.46% | 16.11% | 83.59% | 4.73% | 51.03% | 7.43% |
| Euclidean | All | 18.25% | 19.81% | 59.09% | 5.15% | 37.98% | 10.18% |
| | Good | 4.69% | 21.34% | 39.32% | 3.97% | 18.46% | 8.03% |
| | Mid | 17.19% | 19.55% | 59.44% | 5.46% | 36.12% | 11.07% |
| | Bad | 40.59% | 18.18% | 85.67% | 5.98% | 71.28% | 10.43% |

the users who had at least four valid sessions—password sessions with at least five samples per session—were included in this experiment. With this, the total number of users reduced from 118 to 23 with 2,386 genuine pairs out of the total 9,544 genuine pairs and 2,169 imposter pairs.

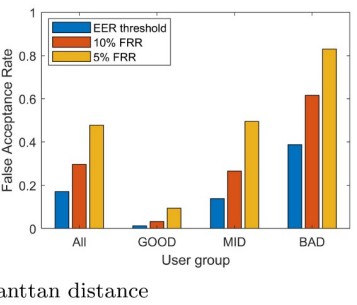 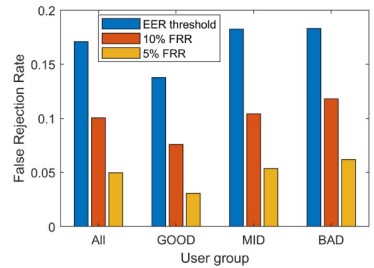

**(a)** Mahanttan distance

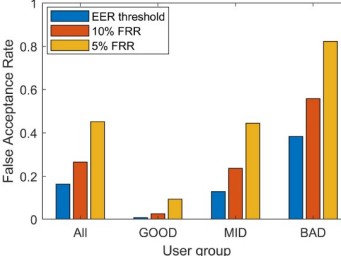 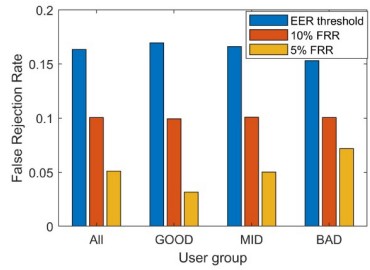

**(b)** Euclidean distance

**Fig 8. Verification performance of free-text username and password combination for each group in terms of FAR and FRR.** Experiments were performed on the Web-based Benchmark dataset based on: (a) Manhattan distance and (b) Euclidean distance.

The verification performance of these username and password combinations is reported in Fig 8. It demonstrates the strong recognition performance for the GOOD users achieving 3.28% FAR at 10% FRR threshold and it is 21 folds lower than those BAD users. Note that, verification performance results in terms of EER for all users are in line with the ones reported in [24].

These results highlight the potential of the proposed distinctiveness metric to be used for constructing a template selection mechanism to enhance the verification performance of the system. That is, keystroke dynamic can be used as an optional secondary authentication factor —it can be ignored or used at choice (of the system) for each user depending on his or her template distinctiveness. Then the system can achieve a strong verification performance at about 5-7% average HTER for GOOD templates compared to 17% HTER for all templates. That is, once template quality can be quantified, it enhances the possibilities for behavioral biometric to be used for validating the identity of the user when adequate template quality is met. As a result, higher security could be provided to those users.

**5.3.4 Performance comparison to existing approaches.** In this subsection, the efficacy of the proposed quality metric for keystroke dynamic templates when compared to existing keystroke dynamic approaches is demonstrated. In addition, the security of keystroke dynamic in comparison to other authentication factors is evaluated.

The performance result of the algorithm used in this paper is reported in Table 5 along with that of existing work. Note that, we have followed the protocol described in [8] to evaluate the overall performance of the algorithm used in this work when compared to existing approaches. That is, the threshold is set individually for each subject and the imposter samples for each subject were from the first five samples of all other subjects. While the reported performance using this protocol can be viewed as the upper bound performance of each algorithm, one

Table 5. Verification performance comparison of existing keystroke dynamic approaches.

| Ref. | Dataset | Number of users | Keystroke length | Algorithm | EER (%) |
|---|---|---|---|---|---|
| [25] | SUNY | 75 | 10 | CNN plus RNN | 9.75 |
| [26] | CMU | 51 | 10 | Dependence Clustering | 7.70 |
| [27] | CMU | 51 | 10 | Nearest Neighborg | 8.40 |
| [8] | CMU | 51 | 10 | Manhattan (scaled with average absolute deviation) | 9.60 |
| This work | CMU | 51 | 10 | Manhattan (scaled with standard deviation) | 9.16 |
| | | | | • Good users | 4.50 |
| | | | | • Mid users | 10.27 |
| | | | | • bad users | 12.70 |

Table 6. Entropy measures of different authentication factors.

| Ref. | Authentication factor | Dataset description | Entropy (bits) |
|---|---|---|---|
| [29] | 4-digits PIN | Dodonew, CSDN, Rockyou, Yahoo (total 3.4 M) | 8.41 |
| [29] | 6-digits PIN | Dodonew, CSDN, Rockyou, Yahoo (total 6.4 M) | 13.21 |
| [30] | Password | 14 datasets (total 113.3 M) | 20-23 |
| [28] | Iris | ICE (High quality set—374 iris, 10 samples each) | 8.9-10<br>8.9-10 bits |
| [31] | Finger Vein | VERA (220 fingers, 2 samples each) | 4.2—13.2 |
| | | UTFVP (360 fingers, 4 images each) | 18.9-19.5 |
| This work | Keystroke | CMU (51 typists, 400 samples each) | 3.48 |
| | | • Good user | 4.62 bits |

challenging issue to be solved when the system is to be used in practical applications is how to set a threshold for each subject to achieve that level of performance. Nevertheless, this result demonstrates that the verification algorithm used in this work is comparable to other existing work when the same dataset and validation protocol is used. In addition, it confirms that the proposed metric can be used to measure quality of the keystroke dynamic template regardless of whether local or global thresholding approaches are used.

Lastly, entropy is another important metric to assess the security level of the system. In Table 6, the entropy measures of different authentication factors are reported. With this information, password-based two-factor authentication is estimated at 23.48-26.48 bits. Note that, the biometric entropy reported in this table is the relative entropy computed from KL divergence between genuine and imposter score distributions [28]. While the entropy of the keystroke dynamic is lower than that of the iris and fingerprint, it offers additional security at no user and hardware cost. That is, users are not required to perform additional tasks and user devices are required to equip with additional hardware.

## 6 Conclusion and future work

In this paper, we proposed a method to derive a quality score of keystroke dynamic templates in terms of distinctiveness. The effectiveness of this proposed score is demonstrated by dividing the keystroke dynamic templates into three equal-size groups according to their distinctiveness scores: the low quality, the middle quality, and the high quality ones. Using the scores computed from the sum of inverse feature variance, the difference between the trade-off curve between the threshold value and corresponding FAR of these two groups of users is also clearly observed. That is, at a range of an operational threshold, the FAR of the templates with higher

distinctiveness is lower than that of the lower distinctiveness ones. The same observation that holds for all three classifiers implies that the proposed distinctiveness score can be used to indicate FAR of the templates across various types of classifiers.

One area of future work is to develop a mechanism to cope with low distinctiveness templates in order to enhance verification performance and reliability of keystroke dynamic verification system. For example, the quality of low distinctiveness templates may be improved intelligently if inconsistent samples are detected and removed. In addition, this paper has demonstrated the efficacy of a derived distinctiveness score when it is used in conjunction with three simple classifiers (Manhattan, Euclidean, and Mahalanobis distance). It is important to investigate the use of the proposed scores when it is applied to more effective or diverse classifiers, e.g., function-based approaches, or feature-based approaches with different sets of features [32], in order to enhance a verification performance of the GOOD users. Also, while the metric is designed to capture the distinctiveness of keystroke dynamic template (both constraint-text and free-text), it can also be applied to other feature-based biometric templates. This could be another fertile research area. Finally, one step towards the practicality of using keystroke dynamic (and other behavioral biometric) as an optional factor for validating the user's identity is to benchmark this quality index using more diverse datasets in terms of the environment of data collection, input device, as well as languages used.

## Author Contributions

**Conceptualization:** Napa Sae-Bae.

**Data curation:** Napa Sae-Bae.

**Funding acquisition:** Napa Sae-Bae.

**Investigation:** Napa Sae-Bae.

**Methodology:** Napa Sae-Bae.

**Software:** Napa Sae-Bae.

**Supervision:** Nasir Memon.

**Validation:** Nasir Memon.

**Writing – original draft:** Napa Sae-Bae.

**Writing – review & editing:** Nasir Memon.

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
