## [Decision Letter · Decision Letter 0]

28 Sep 2021

PONE-D-21-28587Distinguishability of Keystroke Dynamic TemplatePLOS ONE

Dear Dr. Sae-bae,

Thank you for submitting your manuscript to PLOS ONE. After careful consideration, we feel that it has merit but does not fully meet PLOS ONE’s publication criteria as it currently stands. Therefore, we invite you to submit a revised version of the manuscript that addresses the points raised during the review process.

We look forward to receiving your revised manuscript.

Kind regards,

Pandi Vijayakumar, Ph.D

Academic Editor

PLOS ONE

Journal Requirements:

When submitting your revision, we need you to address these additional requirements. 1. Please ensure that your manuscript meets PLOS ONE's style requirements, including those for file naming. The PLOS ONE style templates can be found at https://journals.plos.org/plosone/s/file?id=wjVg/PLOSOne_formatting_sample_main_body.pdf and https://journals.plos.org/plosone/s/file?id=ba62/PLOSOne_formatting_sample_title_authors_affiliations.pdf 2. We suggest you thoroughly copyedit your manuscript for language usage, spelling, and grammar. If you do not know anyone who can help you do this, you may wish to consider employing a professional scientific editing service.  Whilst you may use any professional scientific editing service of your choice, PLOS has partnered with both American Journal Experts (AJE) and Editage to provide discounted services to PLOS authors. Both organizations have experience helping authors meet PLOS guidelines and can provide language editing, translation, manuscript formatting, and figure formatting to ensure your manuscript meets our submission guidelines. To take advantage of our partnership with AJE, visit the AJE website (http://learn.aje.com/plos/) for a 15% discount off AJE services. To take advantage of our partnership with Editage, visit the Editage website (www.editage.com) and enter referral code PLOSEDIT for a 15% discount off Editage services.  If the PLOS editorial team finds any language issues in text that either AJE or Editage has edited, the service provider will re-edit the text for free. Upon resubmission, please provide the following: The name of the colleague or the details of the professional service that edited your manuscript A copy of your manuscript showing your changes by either highlighting them or using track changes (uploaded as a *supporting information* file) A clean copy of the edited manuscript (uploaded as the new *manuscript* file)” 3. Thank you for stating the following in the Acknowledgments Section of your manuscript:  "This work was supported by the grant from National Science and Technology Development Agency (Grant No. FDA-CO-2562-9854-TH). " We note that you have provided funding information that is not currently declared in your Funding Statement. However, funding information should not appear in the Acknowledgments section or other areas of your manuscript. We will only publish funding information present in the Funding Statement section of the online submission form. Please remove any funding-related text from the manuscript and let us know how you would like to update your Funding Statement. Currently, your Funding Statement reads as follows:  "NSFDA-CO-2562-9854-THNational Science and Technology 313Development Agencyhttps://www.nstda.or.th/The funders had no role in study design, data collection and analysis, decision to publish, or preparation of the manuscript" Please include your amended statements within your cover letter; we will change the online submission form on your behalf.

Additional Editor Comments:

The paper needs a major revision.

Reviewers' comments:

Reviewer's Responses to Questions

**Comments to the Author**

1. Is the manuscript technically sound, and do the data support the conclusions?

Reviewer #1: Yes

Reviewer #2: Yes

2. Has the statistical analysis been performed appropriately and rigorously? 

Reviewer #1: Yes

Reviewer #2: Yes

3. Have the authors made all data underlying the findings in their manuscript fully available?

Reviewer #1: Yes

Reviewer #2: Yes

4. Is the manuscript presented in an intelligible fashion and written in standard English?

Reviewer #1: Yes

Reviewer #2: Yes

5. Review Comments to the Author

Reviewer #1: -Abstract is not able to convey what is the technical contribution of this paper. I suggest to re-write it.

-Improve the quality of figures and explain those properly.

-Result section is weak and I suggest authors to add more results and compare those with the existing approaches.

-Although this paper is well written, there are still some typos in the current version. I would like to suggest the authors carefully proofread this paper and correct all the typos in the revision.

-Some very related and recent work may be discussed to improve the quality of literature. e.g.

Efficient and secure routing protocol based on artificial intelligence algorithms with UAV-assisted for vehicular Ad Hoc networks in intelligent transportation systems

An overview of Internet of Things (IoT): Architectural aspects, challenges, and protocols

Security in Internet of Things: issues, challenges, taxonomy, and architecture

Improving Road Safety for Driver Malaise and Sleepiness Behind the Wheel Using Vehicular Cloud Computing and Body Area Networks,

A trust infrastructure based authentication method for clustered vehicular ad hoc networks

The optimal path finding algorithm based on reinforcement learning

The Effect of Gender, Age, and Education on the Adoption of Mobile Government Services

IoT-based Big Data secure management in the Fog over a 6G Wireless Network,

Using vehicles as fog infrastructures for transportation cyber-physical systems (T-CPS): Fog computing for vehicular networks

-The authors are expected to report the running time of the proposed algorithm in the revision.

-As an additional remark, references need to be completed with all the required information (e.g. page number, name of journal/conference, vol., issues, etc).

Based on the comments above, I would like to accept this paper if my following concerns are carefully addressed.

Reviewer #2: This paper addresses the issue of varied security level of Keystroke Dynamic by proposing a metric to quantify the quality of keystroke dynamic templates. Specifically, the metric is derived from statistical information of user specific feature

variations and it can be used to reflect the verification performance of individual

keystroke dynamic template in terms of false acceptance rate (FAR). The results seem nice. Overall, I like this paper, yet there are some issues to be addressed before publication.

Strength:

1) The motivation and idea are good.

2) The comprison of different template groups of keystroke behavior based user authentication in Table is very nice.

3) The experimental results seem to be reseaonable and practical.

Weakness:

1. There is no rationale to explain why the proposed template/metric works. The authors shall use one or two key sentenses to explain why the new template/metric works better than existing schemes.

2. Recently, there have been quite a number of new behavior authentication schemes proposed (as well as other kinds ofauthentication schemes like multi-factor authentication schemes). The authors shall compare the proposed new template with these existing schemes using a table like Table I of the IEEE S&P12 paper and Table IV of the IEEE TII'18 paper.

"The quest to replace passwords- a framework for comparative evaluation of Web authentication schemes", IEEE S&P 2012.

"Measuring Two-Factor Authentication Schemes for Real-Time Data Access in Industrial Wireless Sensor Networks." IEEE TII, 2018.

"Evaluating Behavioral Biometrics for Continuous Authentication： Challenges and Metrics", Proc. Asiaccs 2017.

3. The entropy (security) is also a key factor for biometric/behavior identification. I suggest the

authors compare the entropy of keystroke with other authentication factors. Otherwise, it is difficult

to show the superiority of the keystroke authentication factor. For instance, Sadeghi et. al. showed that EEG

biometric entropy is at best 83-bits; Wang et al. showed that the entropy of human chosen 4-digit PINs is 8.41 bits and 6-digit PINs 13.21 bits; Feng et al. showed that the entropy of human face template is 75 bits; Wang et al. showed that the entropy of human chosen passwords is 20-22 bits; Inthavisas et al. showed that the entropy of voice is 18-30 bits.

Wang et al., Understanding Human-Chosen PINs: Characteristics, Distribution and Security. Proc. ACM ASIACCS 2017.

Sadeghi et al., Geometrical anal- ysis of machine learning security in biometric authentication systems, in:

2017 16th IEEE International Conference on Machine Learning and Ap- plications (ICMLA), 2017

Feng et al., Binary discriminant analysis for generating binary face template, IEEE Transactions on Information Forensics and Security, 2012.

Wang et al., Zipf's law in passwords, IEEE Transactions on Information Forensics and Security, 2017

2776–2791.

Inthavisas et al., Secure speech biometric templates for user authenti- cation, IET Biometrics, 2012.

5. Behavior based user authentication factors are generally not separately used as authentication, especially for behaviors that can be relatively easy recorded/leaked/stealed. Thus, they are often combined with tradtional authentication factors like passwords and PINs.

"Practical and Provably Secure Three-Factor Authentication Protocol Based on Extended Chaotic-Maps for Mobile Lightweight Devices", IEEE Transactions on Dependable and Secure Computing, 2020, Doi: 10.1109/TDSC.2020.3022797

"Unified Biometric Privacy Preserving Three-factor Authentication and Key Agreement for Cloud-assisted Autonomous Vehicles".?IEEE Transactions on Vehicular Technology. DOI: 10.1109/TVT.2020.297125

"Two Birds with One Stone: Two-Factor Authentication with Security Beyond Conventional Bound." IEEE TDSC 2018.

In all, I suggest a “major revision'' to this paper.

6. PLOS authors have the option to publish the peer review history of their article (what does this mean?). If published, this will include your full peer review and any attached files.

Reviewer #1: No

Reviewer #2: No

---

## [Author Response · Author response to Decision Letter 0]

12 Nov 2021

Dear Editor and reviewers,

We are grateful for their critical assessment of our work. These comments have helped us improve our work. We have revised our work according to the recommendation. The concerns are addressed point by point in the response letter as well as in the revised manuscript.

---

## [Decision Letter · Decision Letter 1]

29 Nov 2021

Distinguishability of Keystroke Dynamic Template

PONE-D-21-28587R1

Dear Dr. Sae-bae,

We’re pleased to inform you that your manuscript has been judged scientifically suitable for publication and will be formally accepted for publication once it meets all outstanding technical requirements.

Kind regards,

Pandi Vijayakumar, Ph.D

Academic Editor

PLOS ONE

Additional Editor Comments (optional):

Reviewers' comments:

Reviewer's Responses to Questions

**Comments to the Author**

1. If the authors have adequately addressed your comments raised in a previous round of review and you feel that this manuscript is now acceptable for publication, you may indicate that here to bypass the “Comments to the Author” section, enter your conflict of interest statement in the “Confidential to Editor” section, and submit your "Accept" recommendation.

Reviewer #1: (No Response)

Reviewer #2: All comments have been addressed

2. Is the manuscript technically sound, and do the data support the conclusions?

Reviewer #1: (No Response)

Reviewer #2: Yes

3. Has the statistical analysis been performed appropriately and rigorously? 

Reviewer #1: (No Response)

Reviewer #2: Yes

4. Have the authors made all data underlying the findings in their manuscript fully available?

Reviewer #1: (No Response)

Reviewer #2: (No Response)

5. Is the manuscript presented in an intelligible fashion and written in standard English?

Reviewer #1: (No Response)

Reviewer #2: Yes

6. Review Comments to the Author

Reviewer #1: Distinguishability of Keystroke Dynamic Template is presented in this paper.

Paper is revised well.

Reviewer #2: The revised version has addressed most of my concerns, and I suggest an acceptance.

The experiments are performed based on two public keystroke dynamic datasets

comprising of two main types of keystroke dynamics: constrained-text and free-text,

namely the CMU keystroke dynamics dataset and the Web-Based Benchmark for

keystroke dynamics dataset. Experimental results based on multiple classifiers

demonstrate that the proposed metric can be a good indicator of the template’s false

acceptance rate. Thus, it can be used to enhance the security of the user

authentication system based on keystroke dynamics

7. PLOS authors have the option to publish the peer review history of their article (what does this mean?). If published, this will include your full peer review and any attached files.

Reviewer #1: No

Reviewer #2: No

---

## [Editor Report · Acceptance letter]

3 Dec 2021

PONE-D-21-28587R1 

Distinguishability of Keystroke Dynamic Template 

Dear Dr. Sae-bae:

I'm pleased to inform you that your manuscript has been deemed suitable for publication in PLOS ONE. Congratulations! Your manuscript is now with our production department. 

Kind regards, 

on behalf of

Dr. Pandi Vijayakumar 

Academic Editor

PLOS ONE